# Comparison of Ground-Based, Unmanned Aerial Vehicles and Satellite Remote Sensing Technologies for Monitoring Pasture Biomass on Dairy Farms

**Juan I. Gargiulo** [1,2,*]ⓘD, **Nicolas A. Lyons** [1], **Fernando Masia** [3,4], **Peter Beale** [5], **Juan R. Insua** [4,6], **Martin Correa-Luna** [2] and **Sergio C. Garcia** [2]ⓘD

1  NSW Department of Primary Industries, Menangle, NSW 2568, Australia
2  Dairy Science Group, School of Life and Environmental Sciences, Faculty of Science, The University of Sydney, Camden, NSW 2567, Australia
3  Facultad de Ciencias Agropecuarias, Universidad Nacional de Córdoba, Cordoba 5000, Argentina
4  Consejo Nacional de Investigaciones Científicas y Técnicas (CONICET), Buenos Aires C1425FQB, Argentina
5  Local Land Services Hunter, Taree, NSW 2430, Australia
6  Facultad de Ciencias Agrarias, Universidad Nacional de Mar del Plata, Balcarce 7620, Argentina
*  Correspondence: juan.gargiulo@dpi.nsw.gov.au

**Abstract:** Systematic measurement of pasture biomass (kg DM/ha) is crucial for optimising pasture utilisation and increasing dairy farm profitability. On-farm pasture monitoring can be conducted using various sensors, but calibrations are necessary to convert the measured variable into pasture biomass. In this study, we conducted three experiments in New South Wales (Australia) to evaluate the use of the rising plate meter (RPM), pasture reader (PR), unmanned aerial vehicles (UAV) and satellites as pasture monitoring tools. We tested various calibration methods that can improve the accuracy of the estimations and be implemented more easily on-farm. The results indicate that UAV and satellite-derived reflectance indices (e.g., Normalised Difference Vegetation Index) can be indirectly calibrated with height measurements obtained from an RPM or PR. Height measurements can be then converted into pasture biomass ideally by conducting site-specific sporadic calibrations cuts. For satellites, using the average of the entire paddock, root mean square error (RMSE) = 226 kg DM/ha for kikuyu (*Pennisetum clandestinum* Hochst. ex Chiov) and 347 kg DM/ha for ryegrass (*Lolium multiflorum* L.) is as effective as but easier than matching NDVI pixels with height measurement using a Global Navigation Satellite System (RMSE = 227 kg DM/ha for kikuyu and 406 kg DM/ha for ryegrass). For situations where no satellite images are available for the same date, the average of all images available within a range of up to four days from the day ground measurements were taken could be used (RMSE = 225 kg DM/ha for kikuyu and 402 kg DM/ha for ryegrass). These methodologies aim to develop more practical and easier-to-implement calibrations to improve the accuracy of the predictive models in commercial farms. However, more research is still needed to test these hypotheses under extended periods, locations, and pasture species.

**Keywords:** automation; productivity; calibration; Australia; grazing management

## 1. Introduction

A key driver of farm profitability in pasture-based dairy systems is the amount of pasture utilised or ingested and converted into milk, which is expressed in kg or t per hectare and per year [1–3]. The average pasture utilisation on commercial dairy farms in Australia is ~7 t DM/ha, which is only one-third of the potential achieved under experimental conditions with irrigation [4–6]. Accurate, systematic and timely information on pasture biomass and growth rate is critical to achieving optimum pasture allocation and could increase current milk production by ~10% by utilising pasture that would otherwise be wasted [7]. However, in practice, most dairy farmers only conduct rapid

visual assessments during their daily farm activities instead of using a standardised and regular method to monitor pasture availability [8,9].

Direct cutting and weighing of pasture using a quadrat is widely considered the most accurate method for measuring pasture biomass. However, this approach can be time-consuming, destructive, and requires multiple samples to account for paddock variability [10]. As a result, farmers may prefer to use indirect ground-based methods such as the rising plate meter (RPM) or the electronic pasture reader (PR) as more convenient options [11,12]. Although some regions of Australia and New Zealand report adoption rates of 32–42% for the RPM and 10–11% for the PR [8,13], these figures are likely to be lower on average [9]. The RPM is generally more popular due to its lower cost and high accuracy when used correctly (70 readings in a relatively homogenous paddock are required to achieve a 5% error in the estimations). However, it still requires a significant amount of labour from an operator who must walk the paddocks. For instance, walking a typical dairy farm of around 150 hectares with an RPM could take between 3 and 6 h [14,15], while using a PR could take approximately 1.25 h [16]. Additionally, the collected data often require further processing depending on the device's level of automation, adding to the time and effort required beyond the initial measurement.

The use of unmanned aerial vehicles (UAV) can offer high-resolution data of the entire farm, which can be employed to monitor pasture growth, morphology, digestibility and plant health [17,18]. Although UAVs can reduce the amount of labour required compared to ground-based methods, conducting flights and downloading and processing images can still be time-consuming, especially in large farms. The reliability can also be impacted by weather conditions, especially wind and rain [19,20]. On the other hand, satellite remote sensing is currently seen as the most attractive option to systematically monitor large areas of pasture [21]. Satellites can offer relatively low-cost, high-resolution data with very little associated labour. However, this technology also presents limitations associated with weather conditions (e.g., high cloud cover) or spatial and temporal resolution, which might not be adequate to provide timely and accurate information to farmers [22,23].

In order to achieve accurate measurements of pasture biomass, calibration is necessary regardless of the chosen method or technology. However, the calibration process itself can introduce significant errors [24,25]. Traditionally, local pasture cuts have been used to calibrate these tools, but this method is labour-intensive and time-consuming [10]. Alternatively, standard calibration equations provided by the manufacturer or published in the literature can be used. However, these equations are neither locally developed nor account for factors such as season and pasture species, leading to increased prediction errors [11,26].

Another option, particularly convenient for optical sensors, is to calibrate them indirectly using ground-based measurements obtained from another tool such as the RPM or PR [27]. This indirect calibration approach was successfully evaluated by Flynn [28] for a hand-held device that measured the Normalised Difference Vegetation Index (NDVI), although the performance was lower compared to a calibration against pasture cuts. It is worth noting that the effectiveness of this approach could vary when using satellites or UAVs, as spatial resolution, temporal resolution, and positioning inaccuracies can all impact the measurements. The calibration of satellite-derived data using an RPM was explored in short-term study by Gargiulo et al. [15].

For farmers to adopt pasture monitoring technologies on a large scale, it is important to understand the factors that affect their reliability. This includes evaluating the accuracy of measurements from different sensors (manual, ground-based, aerial, satellite) across various pasture types and seasons. Another critical area of focus is improving calibration methods. Traditional calibration techniques are not feasible for commercial farms, so it is necessary to explore more practical and accessible calibration options. However, no studies have yet examined the differences between direct and indirect calibration methods or the impact of scale (paddock, transect, quadrat) on satellite or UAV-derived data. Our hypothesis is that accurate pasture biomass estimates can be obtained from reflectance

data captured through any sensor, provided they are properly calibrated, and that practical calibration methods can offer sufficient accuracy to promote adoption on commercial farms. To achieve this goal, our study aims to evaluate (i) different ground-based sensors, UAVs, and satellites to monitor pasture biomass accurately as well as (ii) different calibration methods that can be easily implemented on-farm to enhance the accuracy of pasture biomass estimations.

## 2. Materials and Methods

The following section provides a comprehensive description of the three experiments undertaken in this study. Table 1 provides a summary of the experiments, including the general objectives, specific questions addressed and sensors evaluated, as well as the dates and locations of the experiments.

**Table 1.** Summary of the three experiments conducted in this study.

| | Experiment 1 (E1) | Experiment 2 (E2) | Experiment 3 (E3) |
|---|---|---|---|
| Key objectives of the experiment | Calibration methodology | Calibration methodology / Paddock variability evalu-ation | Calibration methodology |
| Platforms and Sensors used [1] | RPM, Automatic PR, UAV [2] | Automatic PR, UAV | C-Dax PR, SAT |
| Reflectance indices evaluated | NDVI, NDRE | NDVI | NDVI |
| Specific questions addressed | Which of the above tools provides more accurate estimates of above-ground pasture biomass at a particular point (i.e., quadrat sites where the pasture was cut and weighted)? | Which of the above tools provides more accurate estimates of above-ground pasture biomass over a given area (i.e., transect)? Can this method be used to estimate inter (and intra) paddock variability? | What are the differences in accuracy between calibration scale (transect or paddock) and pasture species for satellite-based measurements? What are the differences between using images acquired on the same date as the ground measurements versus an average of images available up to +/− 4 days? |
| Duration of the experiment | Short-term (3 weeks) | Medium-term (8 weeks) | Long-term (1 year) |
| Location | Camden (NSW) | Taree (NSW) | Tocal (NSW) |
| Pasture type | Annual ryegrass | Annual ryegrass | Annual ryegrass-Kikuyu |
| Calibration type [3] | Direct | Direct–Indirect | Indirect |
| Calibration Scale [4] | Quadrat | Transect | Transect–Paddock |

[1] RPM = rising plate meter; PR = pasture reader; UAV = unmanned aerial vehicle; SAT = Planet satellite. NDVI = Normalised Difference Vegetation Index, NDRE = Normalised Difference Red-Edge. [2] The UAV was fixed to an arm and used as a ground-based sensor to eliminate errors associated with the positioning system. NSW = New South Wales (Australia). [3] Direct = pasture cuts conducted to calibrate the sensors; Indirect = no pasture cuts conducted, the sensor was calibrated using another sensor (e.g., satellite calibrated using pasture height obtained from a PR). [4] Quadrat = calibrations conducted in a quadrat of 0.25 m$^2$; Transect = calibrations conducted in transects or rows, instead of quadrat; Paddock = calibrations obtained from measurements from the whole paddock.

### 2.1. Experiment 1 (E1)

The aim of this short-term study (three weeks) was to evaluate different calibration methods for estimating pasture biomass at a quadrat scale. Specifically, we compared the accuracy of calibration equations derived between pasture biomass and the measurements obtained from either a RPM, a PR or a UAV (using two vegetation indices, NDVI and NDRE). Direct calibration refers to a sensor calibrated with pasture cuts; quadrat scale refers to measurements taken from a quadrat.

### 2.1.1. Site and Experimental Design

The experiment was conducted from 13 September to 27 September 2018 at The University of Sydney's dairy farm 'Corstorphine', located in Camden, NSW, Australia (34°01′44″S, 150°38′54″E). We used a sector of a paddock sown with annual ryegrass (*Lolium multiflorum* L.) pasture with similar characteristics, covering approximately 1.0 ha (Figure 1).

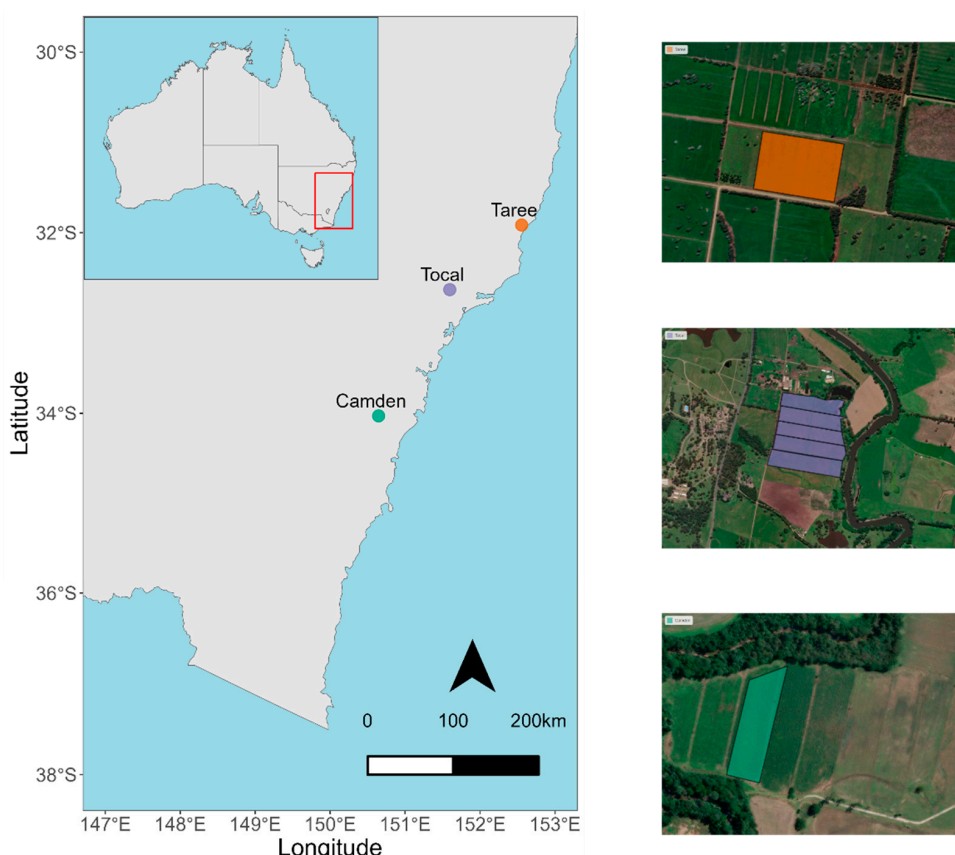

**Figure 1.** Location of the three experiments conducted in New South Wales (NSW), Australia. Experiment 1 (E1) was located at 'Corstorphine' dairy farm in Camden; Experiment 2 (E2) on a commercial dairy farm in Taree, and Experiment 3 at Tocal College dairy farm in Tocal. Maps were created using R software version 4.1.2 (R Foundation, Indianapolis, IN, USA).

### 2.1.2. Data Collection

In this experiment, we selected sectors of the paddock with visually contrasting pasture availability (high, medium, and low) and obtained 12 measurements over three consecutive weeks using a 0.25 m$^2$ quadrat. We used a MicaSense Red Edge camera (https://micasense.com/) (MicaSense, Seattle, WA, USA) mounted on a DJI Phantom 4 UAV (DJI, Shenzhen, China) to obtain multispectral images. This camera senses five spectral bands (red, green, blue, near infrared (NIR) and red-edge). We mounted the UAV on a quad bike with an arm and used it as a ground-based sensor (1.5 m height) to capture a precise image of the quadrat, eliminating any inaccuracies associated with the UAV positioning system. At this height, the spatial resolution of the images is less than 1 mm. To ensure optimal image quality, we followed the manufacturer's recommendations, which included the use of a reflectance panel to compensate for lighting conditions during image capture.

To measure pasture height, we used an 'Automatic' PR device (http://pasturereader.com.au/) (Naroaka Enterprises, Narracan, VIC, Australia), which comprises an ultrasonic sensor mounted on a quad bike. Additionally, we measured compressed pasture height (in 0.5 cm height units) using a Jenquip RPM EC20 (https://jenquip.nz) (Jenquip, Feilding,

New Zealand). The RPM consists of a handle and a plate that slides over a shaft, which is placed over the canopy to measure the height of the compressed pasture material between the plate and the ground. After completing all measurements, pasture material in the quadrat was cut as close as possible to ground level (while avoiding soil contamination), weighed in the paddock, and oven-dried at 80 °C to determine dry matter content and pasture biomass (in kg DM/ha). This methodology was referred to as 'direct calibration at a quadrat scale'.

### 2.1.3. Data Processing and Statistical Analysis

Data processing and statistical analysis were conducted using R software version 4.1.2 (www.r-project.org/). We processed images to calculate the NDVI and the Normalised Difference Red Edge (NDRE) [29,30]. The NDRE is more sensitive than the NDVI at higher biomass levels, and it can only be calculated if the red edge band is available, which is a key feature of the MicaSense camera. For comparative analysis, regression equations were fitted between pasture biomass and the variable measured from each sensor (RPM, PR, and the UAV). The coefficient of determination ($R^2$) was used to test the regressions comparing observations pooled together and per date.

### 2.2. Experiment 2 (E2)

Experiment 2 aimed to evaluate the accuracy of direct and indirect calibration methodologies at a transect scale for the UAV and the PR over eight weeks (medium term). The study also aimed to assess the potential of these methodologies to detect variability in pasture biomass, which was artificially increased by nitrogen fertilisation. Indirect calibration involves using another sensor (such as a PR) to calibrate the UAV. Unlike the quadrat scale used in Experiment 1, the transect scale involved taking measurements from a strip or line.

### 2.2.1. Site and Experimental Design

This experiment was conducted on a commercial dairy farm located in Taree, NSW, Australia (31°53′37″S, 152°34′23″E) from 9 August to 25 September 2018 (Figure 1). We selected a paddock sown with annual ryegrass and assigned three nitrogen fertilisation treatments (30, 60, and 90 kg N/ha) to different sectors of the paddock (Figure 2). The nitrogen fertiliser was applied on the day of grazing (13 August). For the analysis, we used only a homogenous sector of the paddock, covering an area of 4.7 hectares.

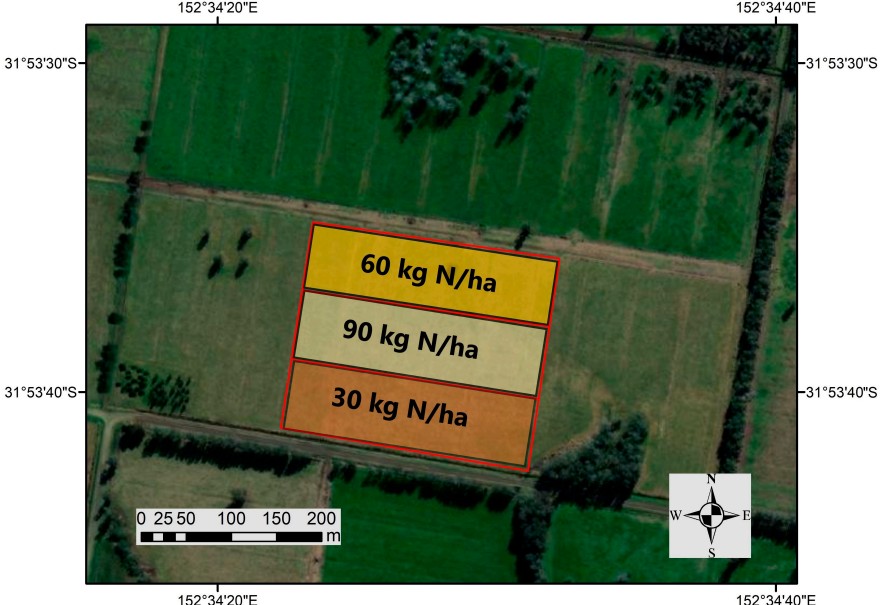

**Figure 2.** Nitrogen fertilisation treatments (30, 60, 90 kg N/ha) evaluated in Experiment 2 (E2) on a commercial dairy farm (Taree, NSW, Australia).

### 2.2.2. Data Collection

We collected pasture height and position data from the entire 4.7 ha paddock, every seven days, using a PR device integrated with a Global Navigation Satellite System (GNSS) receiver. On average, 264 height and location observations were recorded per date. On the same dates, we obtained UAV images using a DJI Phantom 4 mounted with a Sentera NIR camera that captures red, green, blue and NIR spectral bands (https://sentera.com/). The UAV flights were conducted at an altitude of 100 m, with an image overlap of 80% and a spatial resolution of 11 cm. For the calibrations, we measured thirty-two transects of 6.4 m² (13 × 0.49 m) on two dates (14 and 26 September). Each calibration sample involved collecting PR data, extracting UAV reflectance values using the associated PR location, and conducting pasture cuts to calculate DM and pasture biomass. This method was referred to as 'direct calibration at a transect scale'.

### 2.2.3. Data Processing and Statistical Analysis

Images were stitched using Pix4D software (Prilly, Switzerland), and data were processed and analysed using R software. To calculate the NDVI from the UAV images, we followed the instructions provided by the manufacturer. We used direct calibration measurements at a transect scale (Figure 3a) to derive equations between the PR and UAV and pasture biomass. For the PR, we fitted a calibration equation using observations from both calibration dates pooled together, while for the UAV NDVI, we derived two equations (one for each date), since they had a better $R^2$ value (refer to figures in the results in Section 3.2.1). In addition, we tested an indirect calibration methodology at a transect scale for the UAV, which was based on the NDVI and PR measurements from all eight dates. To do this, we first converted each PR observation to pasture biomass and then extracted the corresponding NDVI value from the images (as shown in Figure 3a). We then developed a single calibration equation based on the average NDVI and biomass values for each date and nitrogen fertilisation treatment (i.e., a total of 24 calibration points).

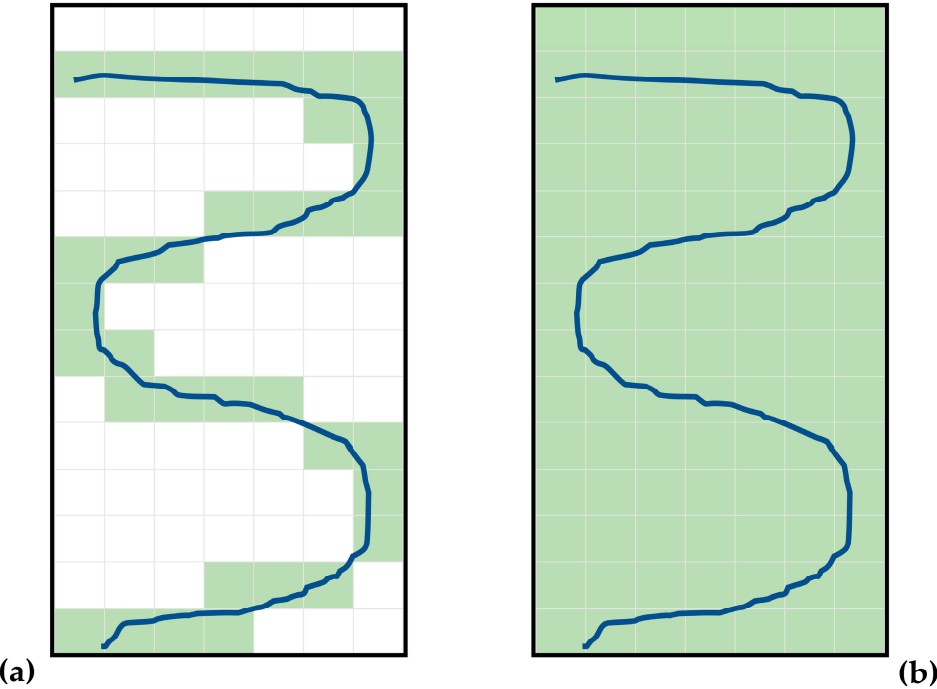

**(a)**    **(b)**

**Figure 3.** Schematic representation of the indirect calibration methodologies used in Experiments 2 and 3. Figure (**a**) ('transect' scale) uses the NDVI extracted from the pixels (green squares) associated with the pattern the pasture reader did on each paddock (blue line). Figure (**b**) ('paddock' scale) uses the average NDVI of the whole paddock.

Moreover, we utilised the obtained calibration equations to transform all PR height and NDVI observations (gathered from the eight dates and each treatment) into pasture biomass. We used these data to construct cumulative pasture biomass curves for each treatment and calibration methodology. Subsequently, we employed a linear model with time and treatment as factors to assess the sensors' capacity to identify differences in pasture biomass between treatments. The interaction between week and treatment was also analysed. We performed these models for NDVI, height, and pasture biomass estimated using all calibration methods presented, including both direct and indirect calibrations at a transect scale.

### 2.3. Experiment 3 (E3)

The primary goal of this long-term experiment (1 year) was to evaluate the accuracy of an indirect calibration methodology at a transect and paddock scale (satellite-derived data) for different pasture species. Paddock scale refers to measurements obtained from the entire paddock area. Additionally, the experiment aimed to evaluate the differences between using satellite images acquired on the same date of ground measurement versus utilising the average of images available up to four days of acquisition date.

#### 2.3.1. Site and Experimental Design

We conducted this experiment at the Tocal College Dairy Farm in Tocal, NSW, Australia (32°37′58″S, 151°35′57″E) between January 2020 and January 2021 (Figure 1). This farm primarily grows kikuyu (*Pennisetum clandestinum* Hochst. ex Chiov) and oversows it with a short rotation ryegrass each autumn. For the experiment, we selected six irrigated paddocks, each covering 4.8 hectares, resulting in a total study area of 29 hectares.

#### 2.3.2. Data Collection

Weekly pasture height measurements were collected using a 'C-Dax' PR (http://www.c-dax.co.nz) towed by a quad bike equipped with a GNSS system to record location information. The 'C-Dax' PR uses a two-sided sensor with light beam emitters to detect relative height when any light paths between the two sides are interrupted. On average, we collected 520 PR height and location observations per paddock per date. In this experiment, we used satellite imagery obtained from Planet Labs Inc. (San Francisco, CA, USA) through the Planet's Education Research Program [31]. We used 'PlanetScope Analytic Ortho Tile' images that include four spectral bands (red, green, blue, NIR) with a spatial resolution of 3.7 m and a revisit time of 1 or 2 days subject to variation based on location and atmospheric conditions [32]. In this study, a total of 75 images were cloud-free and available for use, representing an average of 1 image every 5.2 days.

#### 2.3.3. Data Processing and Statistical Analysis

We tested various calibration methodologies for both kikuyu and ryegrass pastures using the PR and satellite-derived NDVI data. Unlike E1 and E2, this experiment did not involve direct calibration cuts. Instead, we indirectly calibrated the NDVI using the PR height. To achieve this, we utilised the standard equation (kg DM/ha = pasture height × 18.6 + 750) provided by the manufacturer to convert pasture height into biomass [33]. To develop the indirect calibration curves, we first selected satellite images that had PR readings on the same date of acquisition (0 d). Then, we performed the first comparison using either the NDVI values extracted from the pixels associated with the PR position (transect scale) (Figure 3a) or the NDVI and the average biomass values for the whole paddock (paddock scale) (Figure 3b). Since acquiring images on the exact date of PR measurements could be difficult due to temporal resolution or atmospheric conditions, we compared each method (i.e., indirect calibration at a transect or paddock scale, 0 d) with calibration curves derived from using the average of all satellite images available within ± one (1 d), two (2 d), three (3 d) and four (4 d) days from the PR measurement.

## 3. Results

### 3.1. Experiment 1

The direct calibrations between pasture biomass (kg DM/ha) and NDVI, NDRE, height (using a PR), or compressed height (using a RPM) at a quadrat scale are presented in Figure 4. The RPM demonstrated a stronger association with pasture biomass compared to the PR, NDVI, and NDRE ($R^2$ 0.86, 0.64, 0.54, 0.44, respectively). The PR, NDVI, and NDRE had moderate to high $R^2$ values (above 0.70), and regressions were statistically significant ($p < 0.05$) for the first two sampling dates, but for the last date, the $R^2$ was relatively low and the regressions were not significant ($p > 0.05$). Overall, the NDVI exhibited a higher $R^2$ than the NDRE across all evaluated dates.

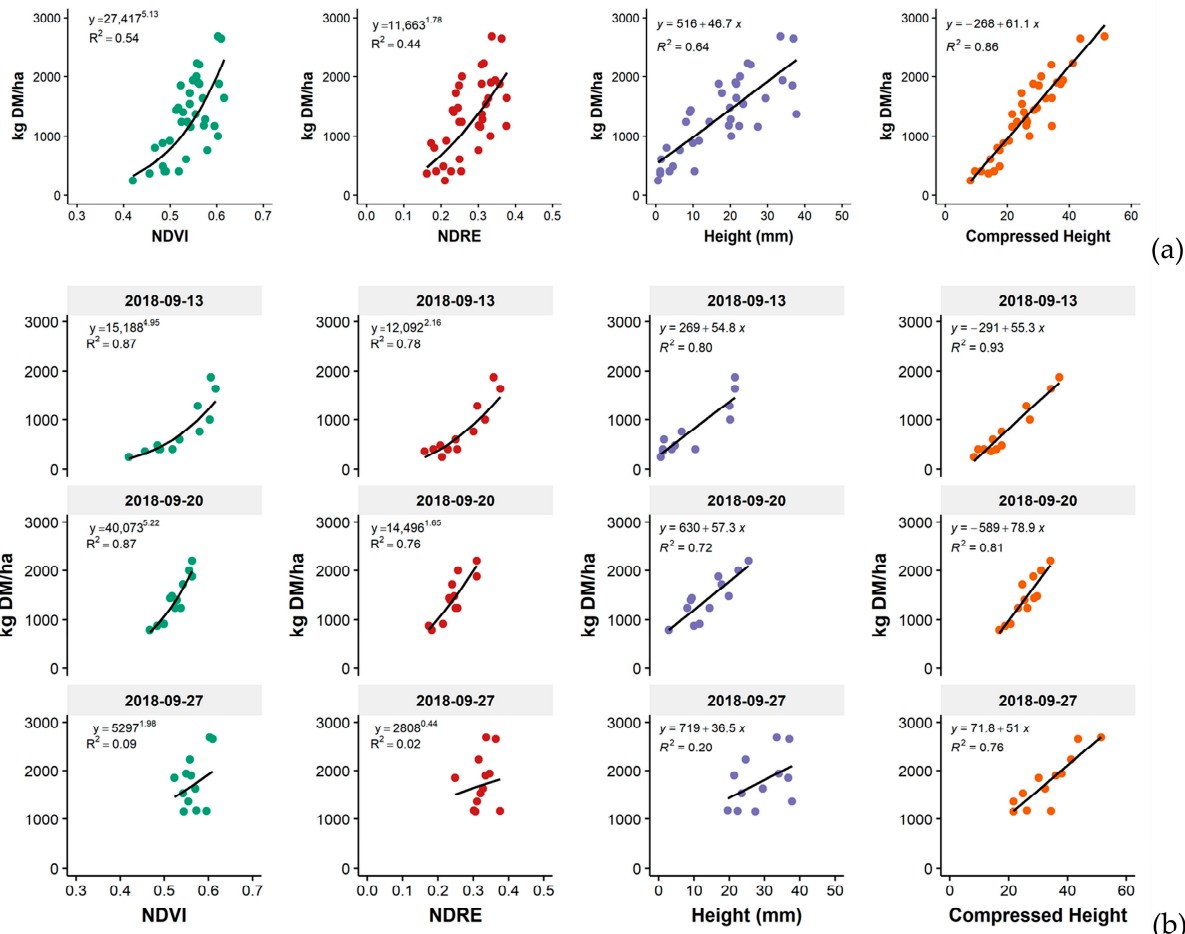

**Figure 4.** Calibration equations for converting Normalised Difference Vegetation Index (NDVI) (●), Normalised Difference Red Edge (NDRE) (●), pasture reader (PR) height (●), or rising plate meter (RPM) compressed height (●) into pasture biomass (kg DM/ha). In (**a**), observations were polled together and in (**b**), observations are shown per date.

### 3.2. Experiment 2

#### 3.2.1. Calibrations

Figure 5a displays the direct calibrations at a transect scale between pasture biomass and PR height and Figure 5b between pasture biomass and NDVI. The two dates were combined into a single linear regression for the PR due to a higher $R^2$, while two separate exponential regressions were kept for the NDVI. The variability explained by the UAV was lower for these calibrations ($R^2$ 0.57 and 0.38) compared to the PR ($R^2$ 0.62). Figure 5c illustrates the indirect calibration for the UAV, which explains a similar variability as the direct calibration for the PR ($R^2$ 0.62).

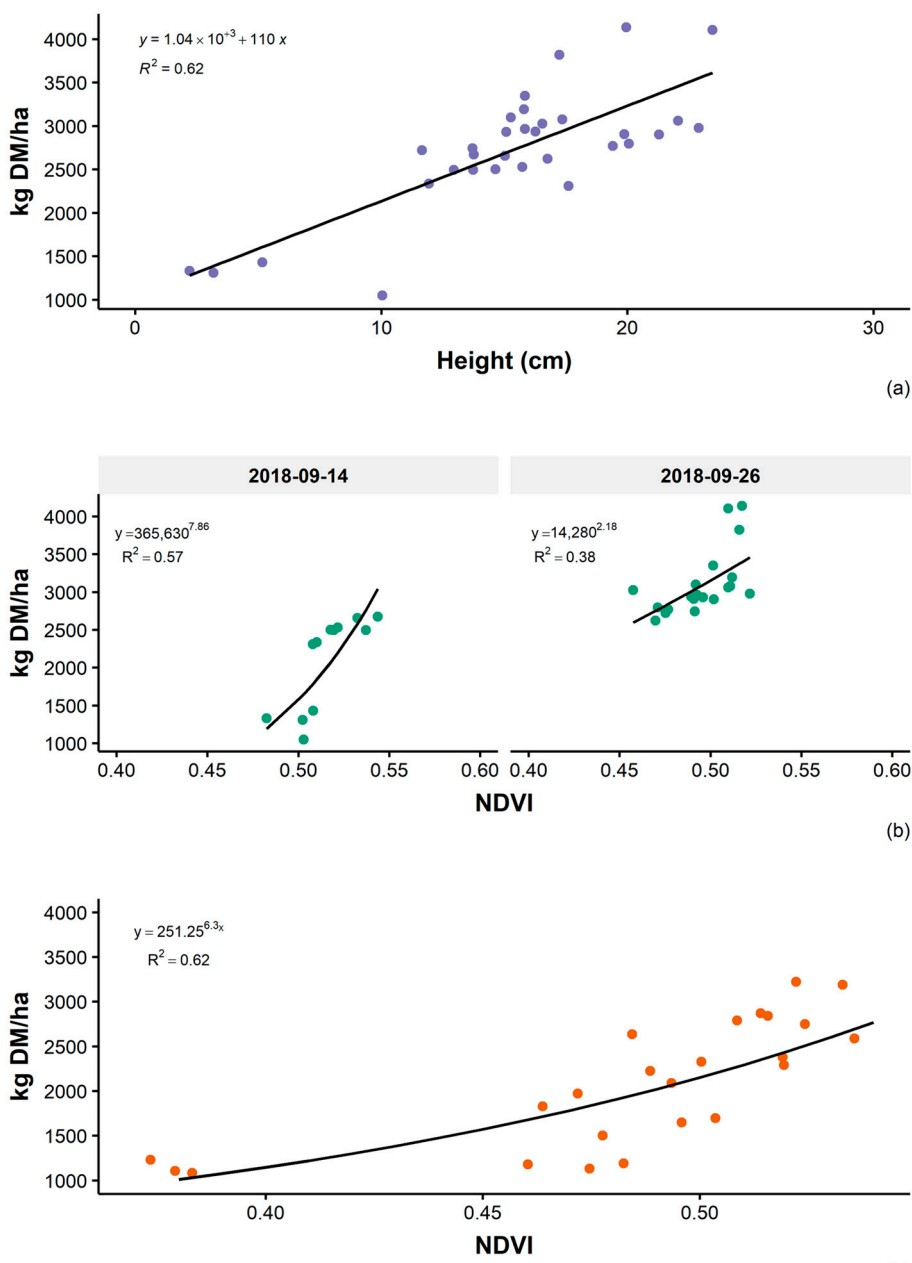

**Figure 5.** Equations for direct calibration of pasture reader height (**a**) (•), and Normalised Difference Vegetation Index (NDVI) (**b**) (•) for conversion into pasture biomass (kg DM/ha). (**c**) (•) presents the indirect calibration equation for converting NDVI into pasture biomass, which involves calibrating NDVI using a calibrated PR.

### 3.2.2. Variability between Treatments

The variability between fertilisation treatments (30, 60, and 90 kg N/ha) was analysed using different methodologies, and the results are presented in Figure 6. The statistical analysis showed that the interaction between date and treatment was significant ($p < 0.05$) for NDVI, height, and pasture biomass estimated using different calibration methodologies. This indicates that the curves representing the treatments were different in all the analyses presented.

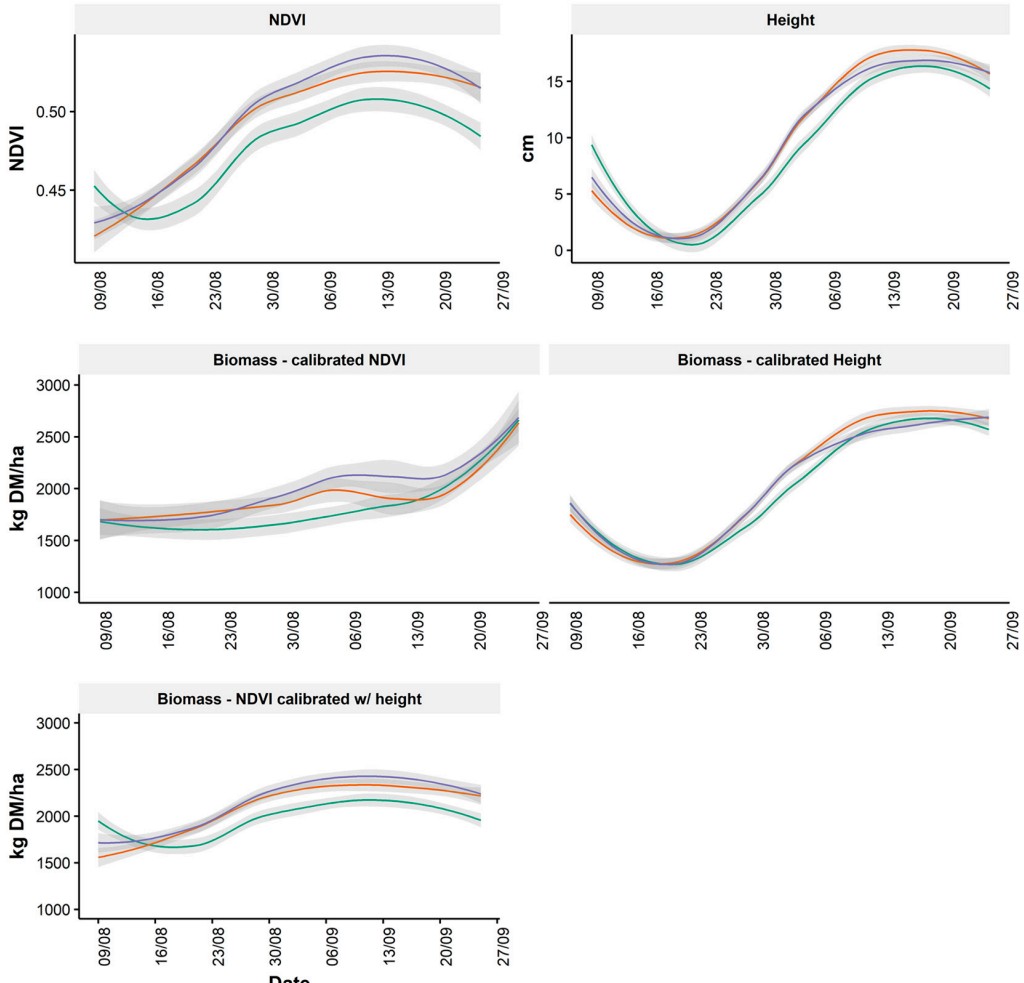

**Figure 6.** Average NDVI, height and pasture biomass (kg DM/ha) estimated using NDVI or height per date. Coloured lines indicate different nitrogen fertilisation treatments; 30 kg N/ha (—); 60 kg N/ha (—); 90 kg N/ha (—). The curves were smoothed using locally estimated scatterplot smoothing (LOESS), and the grey area around the curves represents the 95% confidence interval (CI).

### 3.3. Experiment 3

Exponential relationships between pasture biomass obtained from a PR and from NDVI (Planet satellite) for kikuyu and ryegrass pastures were analysed at the transect and paddock scale (Figures 7 and 8). The calibration curves for kikuyu had a higher accuracy ($R^2 > 0.86$) than for annual ryegrass ($R^2$ ranging from 0.24 to 0.51). The comparison of NDVI values extracted with the PR position (transect scale) and the average NDVI and pasture biomass of the entire paddock (paddock scale) provided similar results. The RMSE and $R^2$ values were also similar when comparing the use of satellite images acquired on the same date of the PR measurement (0 d) versus the average of the images available ±1, 2, 3 and 4 days (1 d, 2 d, 3 d, 4 d) from the PR measurement. This was similar either for kikuyu and ryegrass and for the transect and paddock scale methodologies. The RMSE for 0 d and kikuyu was ~226 kg DM/ha, and the $R^2$ was 0.89 and 0.88 (transect and paddock scale, respectively); for 0 d and ryegrass, the RMSE was 346 and 405 kg DM/ha and the R2 was 0.42 and 0.38 (transect and paddock scale, respectively).

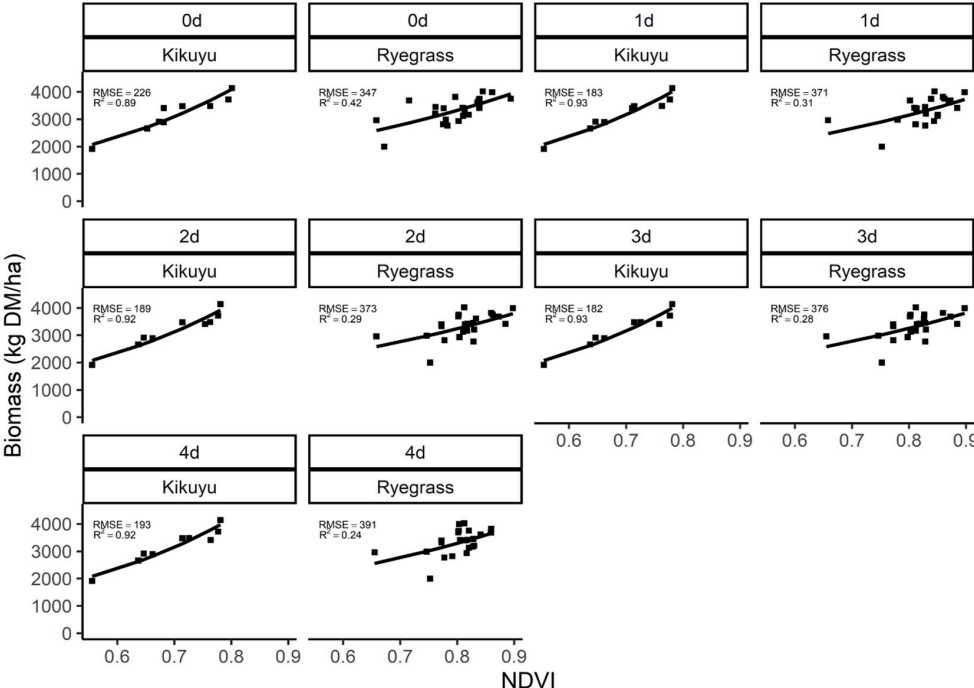

**Figure 7.** Indirect calibration of Planet satellite at a transect scale for ryegrass and kikuyu. The transect scale refers to using the NDVI values obtained from the pixels associated with the drive path taken by the PR (Figure 3a). The calibration was performed using the average of all available satellite images on the same date as the PR measurement (0 d) as well as images acquired within one, two, three, and four days (1 d, 2 d, 3 d, 4 d).

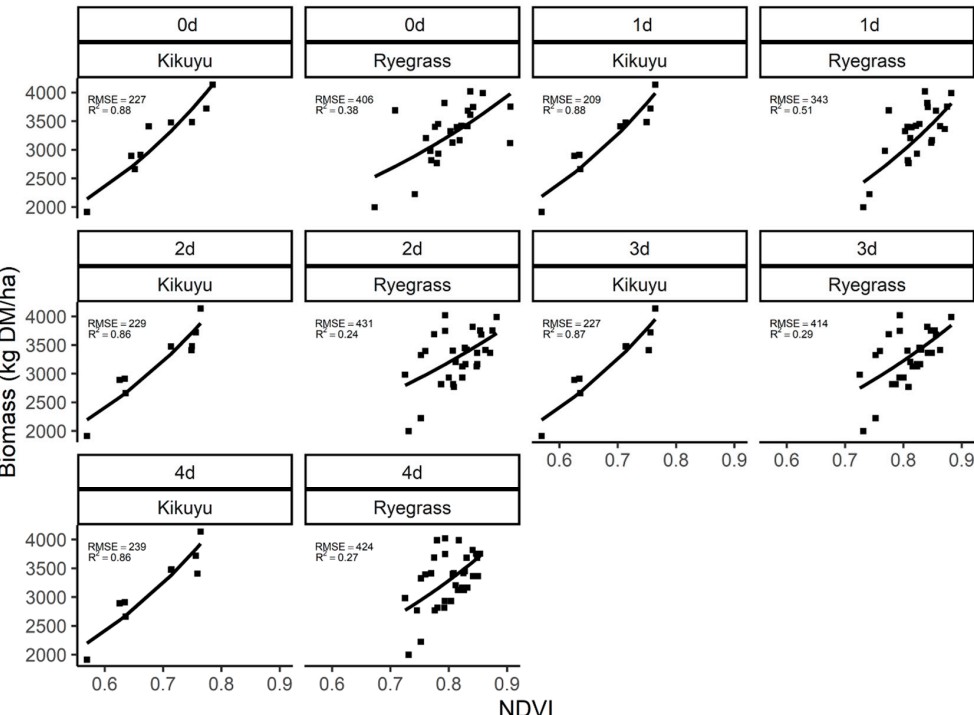

**Figure 8.** Indirect calibration of Planet satellite at a paddock scale for ryegrass and kikuyu. The NDVI values used for the paddock scale were obtained from the pixels of the entire paddock as shown in Figure 3b. The calibration was performed using the average of all available satellite images on the same date as the PR measurement (0 d) as well as images acquired within one, two, three, and four days (1 d, 2 d, 3 d, 4 d).

## 4. Discussion

In this study, we conducted a series of experiments to evaluate the effectiveness of ground-based sensors, UAVs, and satellites under various growing conditions, using different calibration types (direct and indirect) and scales (quadrat, transect, paddock).

In the first experiment, we employed a direct calibration approach at a quadrat scale to evaluate the ability of PR, RPM, and UAV to estimate pasture biomass. While this methodology presented challenges for on-farm practicality and broader applicability, measuring a quadrat by fixing the UAV to an arm reduced errors associated with the positioning system [28,34], and it can be considered a gold standard calibration. The sensor used also enabled the calculation of NDRE, which has been shown in previous studies to have greater sensitivity at higher biomass levels than the NDVI [30]. However, in the three dates evaluated, NDRE had a lower correlation with pasture biomass than NDVI. Among the different sensors, reflectance indices and the PR had lower $R^2$ than the RPM, particularly during late phases of pasture growth (when transitioning into reproductive stages), which was also observed in prior studies [11,26,35]. In the case of the PR (which measures pasture height), this could be due to the structure of the plants and the inability to distinguish stem or seed head from leaf [36]. Furthermore, vegetation indices are significantly affected by standing senescent material [37]. Conversely, the RPM measures compressed height, which is a combination of pasture height and biomass. Figure 4 also illustrates that a single equation for calibrating NDVI might not achieve high accuracy if traditional regression models are used to estimate biomass, as also noted by Gargiulo et al. [15]. Thus, different calibration curves may be needed during periods of high pasture growth or when the pasture is transitioning to the reproductive stage [38]. However, given the impracticality of applying a direct calibration at a quadrat scale on-farm, other methodologies with similar potential accuracy, but easier implementation should be considered instead.

The second experiment aimed to evaluate more practical calibration methodologies at a transect scale for the PR and UAV, and to compare the detection of differences in pasture biomass between nitrogen treatments. In contrast to the first experiment, this was conducted on a commercial dairy farm over a more extended period. The PR calibration showed similar results to E1. However, the direct NDVI calibration method using pasture cuts was less effective in E2 ($R^2$ of 0.57 and 0.38 for E2 compared to an $R^2$ of 0.87 for the first two calibrations of E1). This difference may be due to the larger scale of measurement and inaccuracies in the positioning system used in E2 in contrast to E1 where the UAV was fixed with an arm to a quad bike [34]. On the other hand, the indirect method (i.e., extracting NDVI from the paddock and calibrating it using the calibrated PR) showed a greater association and was similar to that obtained with the PR. Averaging a whole paddock reduced errors associated with the positioning system, making the methodology easier to implement by only requiring the measurement of the average height of the paddock. However, since the NDVI is calibrated indirectly, errors in the PR calibration will add up, so achieving a good initial conversion of height into pasture biomass is essential [28,39–41].

Assessing variability within and between paddocks is crucial in determining the effectiveness of any calibration method. In experiment 2 (E2), we demonstrated that both PR height and NDVI were effective in detecting such variability with respect to biomass. We observed that after the date of nitrogen application (13 August), the average NDVI and height increased with pasture growth, and differences between treatments of 30 kg N/ha and 60–90 kg N/ha were detectable at each measurement. These findings are consistent with previous studies that have used satellite-based NDVI and other reflectance indices to identify differences in pasture chlorophyll, nitrogen content, and biomass [42–44]. When converting NDVI and height curves into pasture biomass using the calibrations, we found that the indirect NDVI method detected differences in pasture biomass more accurately than the other methods (Figure 6). The difference in the type of curve used to convert NDVI or height into pasture biomass may partially explain this result. The exponential curves employed for the UAV calibrated indirectly show that changes in NDVI values (mainly at higher levels of pasture biomass) produced greater differences in pasture biomass than the

linear equation used for the PR [45]. These findings emphasise the importance of selecting an appropriate calibration function, considering not only model accuracy metrics such as RMSE or $R^2$ but also the type of curve utilised [46,47].

On the other hand, Experiment 3 aimed to test the effectiveness of indirect calibrations of NDVI (Planet satellite) with pasture biomass derived from height (C-Dax PR). The experiment was conducted at a larger scale (paddock) over a year and with different pasture species (kikuyu and ryegrass). The calibrations using satellite images on the same date of the ground measurement (0 d) achieved a better accuracy (RMSE of 226–405 kg DM/ha) compared to previous studies conducted in Australia using indirect calibrations and machine learning models. For instance, Asher et al. [48] reported an RMSE between 374 and 610 kg DM/ha for different farms using Planet satellites, while Chen [49] reported RMSE between 324 and 655 kg DM/ha using Sentinel-2 data (lower resolution satellite). Surprisingly, we found that the calibrations for kikuyu were more accurate than for ryegrass, despite the former having a higher proportion of stem and dead material, which can impact the calibrations [50]. It is important to mention that the perceived accuracy of the measurement method is crucial for its adoption by farmers, as many of them would not adopt a method with differences greater than 300 kg DM/ha. Hence, any alternative to the RPM or PR will require farmers' acceptance and their perception of accuracy. Acknowledging this aspect is important for the widespread adoption of satellite technology [27].

Moreover, the results of Experiment 3 indicated that calibrations using either images acquired on the same date as the ground measurements or an average of images within ±four days of the ground measurements produced similar outcomes. It can be challenging to obtain an image for the exact date of ground measurements due to satellite temporal resolution or atmospheric conditions, such as cloud cover or light [21]. However, using the average of multiple images expands the possibilities of more observations available for the calibrations. This aligns with the methodology used by Mata et al. [51] in New Zealand, where the RPM measurements were limited to ±4 days of the image acquisition, achieving a relatively good accuracy ($R^2$ of 0.71 and a residual standard error of 260 kg DM/ha). Additionally, we found no differences between using either the average value for the whole paddock or the image pixel matched to the associated height reading (if the transect and pixel selection is representative of the paddock). Mata et al. [51] calibrated the satellite NDVI at a pixel level, but to achieve between 10 and 20 readings per pixel, an operator walked four transects of 60 m and 15 m apart, taking readings every 1 m (which requires significant effort and time). In that experiment, the operator used a GNSS with a positional accuracy of ±5 m, introducing another source of error to the measurements. The advantage of using the average of the whole paddock is threefold: it avoids the impact of misalignments of the positioning system of the satellite and the PR, is easier to implement, and allows the use of devices that may be already available on-farm, such as PR or RPM with no GNSS.

There are some limitations that need to be considered in interpreting the results of this study. Firstly, the experiments were conducted over different time periods, with E3 being carried out at a later time than E1 and E2. This disparity in timing may prevent a direct comparison of the methods' performance in estimating pasture biomass. Additionally, two of the experiments focused on areas of less than 5 hectares, leading to limited observations and potential variability in the results. This may restrict the generalizability and applicability of the findings to other contexts. Moreover, the study mainly concentrated on single-species pastures, whereas the prevalence of multispecies pastures in dairy farming necessitates further investigation into the implementation of this methodology in such settings. It is likely that the proportion of different species in the mix may be an important factor to account for in future models [52,53].

Devices used for measuring pasture biomass can collect data with varying levels of frequency and accuracy. While traditional regression models can achieve high accuracy, they require site-specific calibrations and may not always be practical due to time constraints or other limitations [15,49]. In contrast, our study aimed to improve the accuracy

of pasture monitoring while significantly reducing labour input by applying more practical and easier-to-implement calibration methods. In this regard, we found that the calibrated RPM could achieve high accuracy but requires significant labour input. The PR could reduce the time required to monitor the farm, but factors such as the reproductive stage of the pasture can affect the accuracy. The UAV has a higher resolution than the satellite and provides the opportunity of producing 3D models of the paddocks (through photogrammetry, Structure-from-Motion or LiDAR), to improve estimations of spectral indices [18,54]. However, still, significant time is required to conduct flights and process data and has limitations such as the proximity to airports and battery life. On the other hand, satellites can provide significant labour savings, but calibrations are still required for high accuracy.

In this study, we found that the best approach to achieve high accuracy and labour savings is to indirectly calibrate satellite NDVI from RPM height measurements, using the average of all observations from the whole paddock (providing the paddock is relatively homogenous) (Table 2). Height measurements can then be converted to pasture biomass using a generic equation or, ideally, site-specific calibrations. Our study also demonstrates that it is possible to use the average of several images within up to ±four days of the ground measurements instead of limiting the method to images captured on the exact date. Further research is still needed to determine how often calibrations should be conducted throughout the year. These results demonstrate promising potential for on-farm implementation of these technologies. However, it is crucial to acknowledge that their accuracy is inherently influenced by specific farm management factors. Therefore, it is essential for farmers to be well informed about these limitations in order to make informed decisions regarding the utilisation of the data provided by these technologies.

**Table 2.** Summary results of the three experiments and calibration methodologies conducted in the study to estimate pasture biomass.

| Experiment | Species | Calibration Type [1] | Calibration Scale [2] | Labour Requirement | $R^2$ | | | |
|---|---|---|---|---|---|---|---|---|
| | | | | | **RPM** | **PR** | **UAV** | **SAT** |
| 1 | Ryegrass | Direct | Quadrat | High | 0.86 | 0.64 | 0.54 | |
| 2 | Ryegrass | Direct | Transect | High | | 0.62 | 0.47 [†] | |
| | Ryegrass | Indirect | Transect | Medium | | | 0.62 | |
| 3 | Ryegrass/Kikuyu | Indirect | Transect | Medium | | | | 0.65 * |
| | Ryegrass/Kikuyu | | Paddock | Low | | | | 0.63 * |

[1] Direct = pasture cuts conducted to calibrate the sensors; Indirect = no pasture cuts conducted, the sensor was calibrated using another sensor (satellite or UAV calibrated using pasture height obtained from a PR or RPM); [2] Quadrat = calibrations conducted in quadrat of 0.25 m$^2$; Transect = calibrations conducted in transects or rows, instead of quadrat; Paddock = calibrations obtained from measurements from the whole paddock. [†] Average $R^2$ for the two regression curves presented in Experiment 2; * Average $R^2$ for Ryegrass and Kikuyu in 0 d.

## 5. Conclusions

In this study, we tested various sensors and calibration methodologies to estimate pasture biomass. We found that the most practical calibration method for satellites and UAVs would be to conduct an 'indirect' calibration at a 'paddock' scale. This involves selecting paddocks with contrasting pasture biomass, obtaining the average NDVI and pasture height for each paddock, and then transforming pasture height into biomass using a generic equation or, preferably, a site-specific equation obtained through pasture cuts. The use of the RPM is recommended over the PR for calibrating NDVI due to its accuracy, practicality, and affordability. Calibrations could be conducted sporadically throughout the year and subsequently applied to other paddocks and dates with available images. If the paddock is sampled correctly, the entire paddock's average can be used, and devices with GNSS are not required. Moreover, the average of images taken up to ±4 days of ground measurements can be used instead of relying solely on images taken on the same day. These findings can serve as a basis for conducting further investigations on calibration methodologies for satellite remote sensing in the context of pasture monitoring.

**Author Contributions:** Conceptualization, J.I.G. and S.C.G.; methodology, J.I.G., S.C.G., F.M. and J.R.I.; investigation, J.I.G. and F.M.; data curation, J.I.G. and F.M.; formal analysis, J.I.G. and F.M.; validation J.I.G.; visualization J.I.G.; writing—original draft preparation, J.I.G.; writing—review and editing, J.I.G., S.C.G., J.R.I., F.M., N.A.L., M.C.-L. and P.B.; supervision, S.C.G. and N.A.L.; resources S.C.G. and P.B. All authors have read and agreed to the published version of the manuscript.

**Funding:** This research received no external funding.

**Data Availability Statement:** Not applicable.

**Acknowledgments:** The authors thank the farmers and dairy farm staff involved in this study. This research was a collaboration between the University of Sydney's Dairy Research Foundation, Hunter Local Land Services and the NSW Department of Primary Industries, conducted as part of the first author's Ph.D. thesis [55]. The first author was the recipient of a Postgraduate Research Scholarship in Automation and Robotics for Dairy Production supported by the NSW Department of Primary Industries and the University of Sydney's Dairy Research Foundation.

**Conflicts of Interest:** The authors declare no conflict of interest.

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
