# Peer review of "Comparison of Ground-Based, Unmanned Aerial Vehicles and Satellite Remote Sensing Technologies for Monitoring Pasture Biomass on Dairy Farms"

_remotesensing, doi:10.3390/rs15112752_

Round 1
Reviewer 1 Report
Authors conducted three experiments in New South Wales (Australia) to evaluate the use of the rising plate meter (RPM), pasture reader (PR), unmanned aerial vehicles (UAV) for monitoring pasture biomass on dairy farms and compared the performance. But the major flaws in the research is as follows.
1. From the experimental section , it is observed that Experiment 1 (RPM) was conducted during 13 September to 27 September 2018, Experiment 2 (PR) was conducted during 9 August to 25 September 2018 and Experiment 3 SAT was conducted during January 2020 and January 2021.
The experiment 3 was conducted in a different time period as compared with experiment 1 and 2. Hence, we can't compare the performance of the methods to estimate pasture biomass in different period experimentation outcome.
2. Many recent literatures are not included in the introduction part. For example, no single paper from 2022 is included.
3. Research contributions can be highlighted in Introduction section.
4. Discussion part needs to be improved.
5. Limitations of the proposed research should be included.
Good.
Reviewer 2 Report
Review report: remotesensing-2373007
Manuscript Number: remotesensing-2373007
Title: Comparison of ground-based, unmanned aerial vehicles and satellite remote sensing technologies for monitoring pasture biomass on dairy farms
General remarks
The topic of the study is interesting and have significant meaning in the remote sensing sciences and pasture biomass; however, there have some shortcomings in the manuscript, which is need to work more finely before consideration in the journal of Remote Sensing.
Detailed comments
Abstract
Line 27: NDVI. Write it with full form.
Line 30: kikuyu. Define it.
Line 31: ryegrass. Define it.
Line 27: Add the resolution of the data taken by UAV.
Keywords: Add one keyword as the geographical area of the study and limit keywords within the five keywords.
Introduction
First paragraph: The research gap in the first paragraph is not seems fine. I suggest to merge it in the last paragraph and re-write the introductions.
In the introduction section, there are missing to connect most recent literature regards this research issue.
Materials and Method
Figure 1 of the manuscript is not prepared finely, try to make it more attractive. The legend is missing within the map, and it will be fine to show DEM map within the study site. I suggest to make it finer with adding appropriate legend in the map and merge two map within one.
Also suggest to improve figure 2 in the same way.
Data Processing and Statistical Analysis. In this section mentioned figure 5a, 5b and 5c before figure 3. Carefully check and correct it.
Results, Discussion and Conclusion
The analysis of the results, discussion and conclusion are fine.
Minor editing of English language required.
Reviewer 3 Report
The manuscript is well written and the subject falls within the scope of the SI as it deals with 'the comparison of different data sources/types, platforms and spatial, spectral and temporal resolutions for crop and rangeland monitoring' (as mentioned in the description of the SI).
I do have some minor comments:
- L 60-62: Another added value (in my opinion) of UAV imagery here is the collection of information about plant condition / the presence of stress (e.g. through the NDRE-index used in this study)
- Table 1: I am not sure if 'Sensors used' is the best description. I believe that UAVs are not sensors per se, but rather platforms. It might be better to adapt this to 'Platforms and sensors used'.
- L139: Crucial information is missing here. Which type of UAV (model) was used? What was the flying height and ratio of image overlap? As these flight parameters have an influence on the quality of the output, it is best to mention them here.
- L158: Which software was used to stitch the different UAV images?
-L168: Same UAV as in E1?
-L238-241: How many of these images/days were useful (e.g. no clouds)?
- Discussion: In general this discussion is very informative. One element I would include is a discussion on additional UAV applications/sensors. Through photogrammetry or Structure-from-Motion, it is possible to generate a detailed 3D model of a parcel. Such information can be interesting to predict pasture biomass, or further improve estimates based on NDVI. To this end, LiDAR data can be useful as well.
Reviewer 4 Report
Generally very well written paper which is logically put together and well illustrated. The paper needs a few minor edits to the grammar/syntax. There could be some more illustrations re: fields at ground level, drone data acquisition and so on - if there is space. I would like to have seen some more context on the research. As it stands this is a useful paper re: comparisons of how the information needed for decision-making is and can be acquired with some interesting results. However, that said it would be good to understand how much information is actually needed in practice and how the decision-making is made now and what the accuracy needed or thresholds are to see how one or other of these is actually an improvement etc. This would I think provide a useful lead into the research and would provide a useful rationale for the work - at least for the reader. Just an additional paragraph would suffice. I wondered also about 'tests' of the methodologies in a number of separate sites just to see how well each approach works in practice. A minor edit might be the quality of the study area maps which are a little cartographically lacking.
The paper needs a few minor edits to the grammar/syntax. This will easily be accomplished by a final read over of the paper just to tidy it up a little.
Reviewer 5 Report
Overall I enjoyed the paper. It was simple, well presented and insightful. I have very few comments or suggestions.
My main comment is about the lack of observations since most of the research was carried out over very short periods and/or on only one small camp (less than 5 ha). This causes the variability in some of the results, and limits the significance of the results. I do not believe it is a limitation to the research being published, but it limits the insightfulness to be applied into other contexts. Nothing can be done about it at this stage, but I thought it pertinent to mention that. Further to this is the limitations caused by only working on single-species pastures. Multispecies pastures have become widely adopted on dairy farms, and therefore this research is limited to basic insight. This has been acknowledged in the paper, but a direct reference to multispecies pastures, and possible limitations to the implementation of this methodology to such pastures would be beneficial.
I have also found a large difference in the perceived accuracy between statistical significance, and on-farm practical significance. Anything greater than 300 kg DM/ha is a large difference for a farmer. I do not have any direct suggestions for changes that need to be made, other than the possibility of acknowledging that an alternative measurement technique to RPM or PR will require farmers' buy-in, and it is the farmers' perception of accuracy that will cause the widespread adoption, or non-adoption, of satellite readings for pasture biomass measurements.
In light of this comment, I believe that the statement in line 263 and 264, saying that the R2 values for PR, NDVI, and NDRE were high is maybe an overstatement. R2 values below 0.7 do not inspire great confidence for practical application, even when the regression is statistically significant.
I believe that the paper is well-written and well presented.
Round 2
Reviewer 1 Report
It can be accepted.
Minor English corrections are required.
Reviewer 2 Report
The authors improved the manuscript and solved all my concerns, comments and suggestions. Thus, I recommend acceptance of this manuscript for publication in the journal of Remote Sensing.
Minor editing of English language required